# A Preliminary Study of Cu Exposure Effects upon Alzheimer’s Amyloid Pathology

**DOI:** 10.3390/biom10030408

**Published:** 2020-03-06

**Authors:** Alexander Pilozzi, Zhanyang Yu, Isabel Carreras, Kerry Cormier, Dean Hartley, Jack Rogers, Alpaslan Dedeoglu, Xudong Huang

**Affiliations:** 1Neurochemistry Laboratory, Department of Psychiatry, Massachusetts General Hospital and Harvard Medical School, Charlestown, MA 02129, USA; apilozzi@mgh.harvard.edu (A.P.); jack.rogers@mgh.harvard.edu (J.R.); 2Neuroprotection Research Laboratory, Departments of Radiology and Neurology, Massachusetts General Hospital and Harvard Medical School, Charlestown, MA 02129, USA; yu.zhanyang@mgh.harvard.edu; 3Department of Veterans Affairs, VA Medical Center, Bedford, MA 01730, USA; carreras@bu.edu (I.C.); kerry.cormier@va.gov (K.C.); dedeoglu@bu.edu (A.D.); 4Department of Neurology, Boston University School of Medicine, Boston, MA 02118, USA; 5Autism Speaks, Chicago, IL 60659, USA; dean.m.hartley@gmail.com

**Keywords:** Alzheimer’s disease, amyloid precursor protein, Aβ amyloid, copper, cytokine, neuroinflammation

## Abstract

A large body of evidence indicates that dysregulation of cerebral biometals (Fe, Cu, Zn) and their interactions with amyloid precursor protein (APP) and Aβ amyloid may contribute to the Alzheimer’s disease (AD) Aβ amyloid pathology. However, the molecular underpinnings associated with the interactions are still not fully understood. Herein we have further validated the exacerbation of Aβ oligomerization by Cu and H_2_O_2_ in vitro. We have also reported that Cu enhanced APP translations via its 5′ untranslated region (5′UTR) of mRNA in SH-SY5Y cells, and increased Aβ amyloidosis and expression of associated pro-inflammatory cytokines such as MCP-5 in Alzheimer’s APP/PS1 doubly transgenic mice. This preliminary study may further unravel the pathogenic role of Cu in Alzheimer’s Aβ amyloid pathogenesis, warranting further investigation.

## 1. Introduction

More than 5 million Americans are affected by the most common senile dementia, Alzheimer’s disease (AD), and more than $100 billion is spent annually on direct care for AD patients in the US alone. The problem is worsening as life expectancy continues to increase. By 2050, the number of AD patients is projected to exceed 88 million worldwide if no cure or preventive measure for AD is found. Thus, there is an urgent need for effective treatments and prevention measures for AD as it has become a public health hazard [1].

Genetic, biochemical, and neuropathological data strongly suggest that Aβ amyloidogenesis may be one of the key events in AD pathogenesis [2]. Considerable efforts have been expended on studying the basic biology of the amyloid precursor protein (APP) and other AD-related proteins to identify the pathways of Aβ metabolism. However, the environmental and neurochemical factors that promote age-dependent Aβ amyloidosis have received much less attention.

Mounting evidence indicates that dyshomeostases of cerebral biometals such as Fe, Cu, Zn, and APP/Aβ/metal redox interactions, may contribute to the neuropathology of AD [3,4,5,6,7,8]. Studies have confirmed that insoluble Aβ amyloid plaques in post-mortem AD brains and the hippocampus of AD mouse models have abnormal enrichment of Cu, Fe, and Zn [9,10,11]. Conversely, metal chelators can regulate metal toxicity, dissolve these proteinaceous deposits from post-mortem AD brain tissue, and attenuate cerebral Aβ amyloid burden in APP transgenic mouse models of AD [12,13,14]. However, one study suggests that only copper presents a risk for AD, and zinc and iron do not, and there is some controversy surrounding chelation of these metals; the relationship of these metals to AD is obviously complicated [15,16,17,18,19]. Furthermore, we and others have shown that metals promote in vitro aggregation of Aβ into tinctorial amyloid [20,21,22]. Additionally, we have demonstrated that redox-active Cu(II), and to a lesser extent, Fe(III), were reduced in the presence of Aβ with concomitant production of reactive oxygen species (ROS)- hydrogen peroxide (H_2_O_2_) and hydroxyl radical (OH•) [23,24,25]. These Aβ/metal redox reactions, which were silenced by redox-inert Zn(II) [26], but exacerbated by biological reducing agents [27], have engendered Aβ oligomerization in vitro mediated by dityrosine cross-linking [28,29]. The very Aβ/metal redox interactions may directly lead to the widespread oxidation damage observed in AD brains [30] and contribute to AD pathogenesis consequentially. Copper also fuels inflammatory factors in the brain [31], and has been implicated in mitochondrial dysfunction, which is a characteristic of AD, and processes involving the beta-secretase protein (BACE1) [32,33]. It should be noted that other non-essential metals have also been associated with AD, such as mercury [34] and lead [35,36,37], among others [36,37].

In regard to APP, the 5′-untranslated region (5′UTR) of APP mRNA has a functional iron-response element, (IRE) [38] and this finding is consistent with biochemical evidence that APP is also a redox-active metalloprotein [39]. Furthermore, compared to age-matched controls, gene expression levels for metal regulatory proteins such as metallothionein III (MT-III) and metal regulatory factor-1 (MTF-1) were decreased more than 4 fold in the AD brain [40]. Moreover, levels of the MT-III protein were reduced in the AD brain [41,42]. Hence, the emergence of redox-active metals as key players in Alzheimer’s amyloid pathogenesis strongly argues that metal-complexing agents should be investigated as possible disease-modifying agents for treating AD.

In fact, an early study indicated that sustained intramuscular administration of a potent iron chelator, desferrioxamine, (DFO) slowed the clinical progression of AD dementia [43]. More recently, DFO has been shown to slow cognitive decline in transgenic APP/PS1 mice and non-transgenic mice [44,45] and to heighten memory in AD rats [46]. The metal-complexing agent clioquinol (CQ) appeared to improve cognitive health in MCI and AD patients [47,48], though further studies of the drug were discontinued, and evidence to support its efficacy and safety is lacking [49]. Indeed, not much research in the recent years has been devoted to further investigating more metal-chelating agents for AD, so more research is required to evaluate metal-complexing effects upon AD.

The brain is a specialized organ that normally concentrates Cu, Fe and Zn in the neocortex [9]. Recent data indicate that cerebral homeostasis of Cu, Fe, and Zn is closely associated with AD [50,51]. It has been shown that endogenous synaptic zinc contributes to cerebral amyloid deposition in APP2576 transgenic mice lacking zinc transporter 3 [52]. Further, trace amounts of copper in water induce Aβ amyloid plaques and learning deficits in animal models [53,54], and humans taking copper supplements exhibit a faster decline in their cognitive abilities [55]. However, there is some evidence to show that low levels of copper, as opposed to excess copper, are common in AD [56] and there is evidence that both deficiencies in, and excesses of, copper inhibit normal regulation of Aβ [57,58]. APP modulates copper-induced toxicity and oxidative stress in primary neuronal cultures [59]. Copper depletion down-regulates APP expression [60] while cellular copper levels are increased in primary mouse cortical neurons and embryonic fibroblasts from APP gene knock-out mice [61]. However, both dietary Cu exposure and endogenous Cu elevation reduces the Aβ burden in vivo [62,63,64,65], and one study suggests that copper depletion in AD is not widespread [66]. One might expect that countries utilizing copper plumbing would experience greater incidences of AD than those that do not if the copper/AD hypothesis holds true, but results are inconclusive [54,67]. These data suggest a complex role of Cu in APP metallobiology [68].

## 2. Materials and Methods

### 2.1. Effects of Cu and H_2_O_2_ upon Aβ oligomerization In Vitro

Aβ1-40 (synthesized at Brigham and Women’s Hospital Biopolymer Laboratory) was dissolved in 6 M Guanidine HCl at 1 mg/ml, spun at 13,000× g for 10 min and purified by size-exclusion chromatography (SEC) isocratically using a Superdex G75 column (GE Healthcare, Chicago, IL, USA) and 70 mM NaCl + 5 mM Tris buffer, pH7.4. Aβ was eluted as a single peak and during purification was exchanged into the SEC buffer (i.e., removing the guanidine HCl). Experiments in western blot represents 11.3 μM purified Aβ alone or with 50 μM CuCl_2_ and/or 250 μM H_2_O_2_ at 37 °C for 3 h. After incubation period samples were spun 3 times through a 3 kD MW cutoff filter using the SEC buffer as a wash buffer to remove Cu and H_2_O_2_. Western blot represents 20 μL of reaction mixture in which 1× loading buffer was added and sodium dodecyl sulfate polyacrylamide gel electrophoresis (SDS-PAGE) was performed on a 16% Tris-tricine gel (Invitrogen, Carlsbad, CA, USA). Bands were visualized using anti-Aβ antibody (6E10, Signet Laboratories Inc., Dedham, MA, USA), HRP-conjugated secondary antibody (Jackson ImmunoResearch, West Grove, PA, USA) and a chemiluminescent detection method per manufactures protocol (ECL Plus, GE Healthcare, Chicago, IL, USA).

### 2.2. Effects of Cu Treatment Upon APP Expression

#### 2.2.1. Effects of Cu Treatment Upon APP mRNA 5′UTR Translation

To evaluate the effects of Cu^2+^ on APP5′UTR translation, human SH-SY5Y neuroblastoma cells, transfected stably with pIRES (APP mRNA 5′UTR driven luciferase reporter gene construct with GFP as an internal specificity control), were purchased from the ATCC (Manassas, VA, USA) and were used following our recently published procedure [69]. In brief, the cells were trypsinized, harvested, resuspended in complete growth medium (DMEM supplemented with 10% FBS, 1% Penn-Strep, and 300 μg/mL of geneticin) and strained with a cell strainer (BD Falcon, Bedford, MA, USA) to remove clumped cells and ensure a more equal number of cells in each well. A Multidrop 384 cell dispenser (Thermo Lab systems, NJ, USA) was used to add 40 μL containing 1000 cells to the 23 columns of black 384-well culture plates. Untransfected SH-SY5Y cells were added to the 24th column. The cells were incubated for 2 h before addition of Cu^2+^. The cells were finally incubated with Cu^2+^ (0.01, 0.1, 1 μM) for 48 h under the cell culture conditions (95% O_2_, 5% CO_2_, 85% humidity, 37 °C) before measuring luminescence on an LJL Analyst (Molecular Device, Sunnyvale, CA, USA). The luminescence data was uploaded into Activity Base (IDBS, Inc., Boston, MA, USA), exported to Excel, and the inhibition of luciferase activity was calculated from the luminescence by the formula: % inhibition = 100 × [1 − (luminescence-avg neg. control)/(avg. pos. control- avg. neg. control)]; where avg. neg. control is the average luminescence calculated from wells with DMSO and untransfected SH-SY5Y cells (column 24) and avg. pos. control is the average of luminescence from the positive control wells containing transfected cells with DMSO (column 23).

#### 2.2.2. SDS-PAGE Analysis of Cu Effects on APP Protein Expression in Human SH-SY5Y Neuroblastoma Cells

To assess the effects of Cu^2+^ on endogenous APP protein expression, SH-SY5Y neuroblastoma cells were grown in DMEM supplemented with 10% fetal bovine serum (BioWhittaker™, Walkersville, MD, USA) and the antibiotics. When the cells reached 40–50% confluences, they were exposed to Cu^2+^ at various concentrations in DMEM (0 or 1 μM) under the cell culture conditions (95% O_2_, 5% CO_2_, 85% humidity, 37 °C) for 3 days. After Cu^2+^ treatment, the cells were collected and washed with cold PBS 3 times and then lysed with M-PER™ mammalian protein extraction reagent (Pierce, Waltham, MA, USA) supplemented with protease inhibitor cocktails (Roche, Basel, Switzerland). The cell lysates were pelleted at 13,000 rpm × 15 min in a cold room, and the supernatants were BCA assayed for total protein concentrations (Pierce, Waltham, MA, USA). Western blotting was performed according to the procedure as previously described.

### 2.3. Effects of Dietary Cu Exposure on Cerebral Aβ Amyloid Pathology in APP/PS1 Transgenic Mice

#### 2.3.1. Treatment of APP/PS1 Transgenic Mice by Cu and Preparation of Mouse Brain Tissue Sections

APPTg2576x PS1(M146V) (APP/PS1) transgenic female mice, approximately 6-month-old, were used for the pilot in vivo studies. The animal research protocol (Protocol #: 2017N000140) has been reviewed and approved on July 7, 2017, by the Institutional Animal Care and Use Committee (IACUC – OLAW Assurance # D16-00361) of Massachusetts General Hospital. The protocol as submitted and reviewed conforms to the USDA Animal Welfare Act, PHS Policy on Humane Care and Use of Laboratory Animals, the “ILAR Guide for the Care and Use of Laboratory Animals” and other applicable laws and regulations. This protocol is approved for a three-year period, subject to submission of annual reports. The mice were genotyped for both APP and PS1 transgenes before the experiments. The mice were treated with regular (3 mice) and Cu-enhanced animal diets (5 mice) ordered from Purina Test Diets, Richmond, IN, USA, for 24 days. The mice on Cu-enhanced diets received approximately 300 mg Cu/kg daily dose. The transgenic mice were closely monitored for general well-being such as body weight change, food consumption rate, and gastrointestinal side effects. At the end of Cu treatment, mice were deeply anesthetized and transcardially perfused with PBS; cortical and striatal tissues were then dissected from each mouse, placed in Eppendorf tubes, frozen in dry ice, and stored in a −80 °C freezer for further experiments. Half of them were used for immunohistochemical (IHC) studies, and the other half were reserved for other biochemical and metal analyses. Immunohistochemical staining and semi-quantification of cerebral Aβ amyloid in APP/PS1 transgenic mice.

The cortical tissues used for immunohistochemical (IHC) staining were post-fixed with the Periodate-Lysine-Paraformaldehyde solution for 24 h, weighed, and cryoprotected in a graded series of 10 and 20% glycerol/2% DMSO solution. Brains were then serially cut at 50 µm thickness on a freezing microtome and stained with cresyl violet to identify histopathological abnormalities. Sections were immunostained for Aβ1-42, Aβ1-40, glial fibrillary acidic protein (GFAP) and synaptophysin (SYP) as markers for Aβ deposits, reactive astrocytes, and synapses. Immunohistochemical procedures were performed as previously described [70]. In brief, free-floating sections were incubated overnight in primary antibody solutions followed by PBS washes and incubation in solutions of peroxidase-conjugated secondary antibody. The sections were developed using DAB as a chromagen. The following primary antibodies were used in immunostaining for brain sections of the transgenic mice: anti-Aβ1-42 (Cat#44-344, BioSource International, Camarillo, CA, USA), anti-Aβ1-40 (Cat#44-348, BioSource International, Camarillo, CA, USA), anti-GFAP (MAB3402, Chemicon), and anti-SYP (Cat#17750, Santa Cruz Biotechnology, Dallas, TX, USA).

To semi-quantify these IHC data, three serial sections per mouse brain were analyzed blindly using Stereo Investigator v. 6.55 (MicroBrightField Inc., Williston, VT, USA) at 40× magnification. The most rostral section analyzed was at the anterior commissure level (0.1 mm anterior to bregma) and each successive section was at 0.3 mm increments caudal to the first. In each section the cortex was manually traced and then plaques over a threshold diameter of 7.6 μm were traced automatically after manual verification. Cortex area, total plaque area, plaque area fraction, and plaque count were then exported to Microsoft Excel for each section using NeuKPLrolucida Explorer v. 4.50.4 (MicroBrightField Inc., Williston, VT, USA) and custom software. The mean and standard error for these parameters from these mouse brain tissue sections were then calculated using Microsoft^®^ Excel for Mac 2019 (Microsoft, Redmond, WA, USA).

#### 2.3.2. Cerebral Aβ Measurements by ELISA

To prepare tissue lysates, frozen cortical tissues in the tubes were thawed on ice. Cold T-PER™ tissue protein extraction solution (Pierce, Waltham, MA, USA) supplemented with Complete™ protease inhibitor cocktail (Roche, Basel, Switzerland) were added to the tubes by 500 µL aliquots and homogenized thoroughly (1800 rpm) with a micro-tube pestle (Research Products International Corp., Mt. Prospect, IL, USA) mounted onto an electric motor (IKA Labortechnik, Staufen, Germany) in the cold room. Homogenates were centrifuged at 13,000 rpm × 15 min in the cold room. Supernatants were carefully decanted, BCA-assayed for total protein concentrations (Pierce, Waltham, MA, USA), and stored in a −20 °C freezer for further analyses. Colormetric ELISA for both Aβ1-40 and Aβ1-42 in the supernatants was performed using commercial kits.

#### 2.3.3. SDS-PAGE Analysis of Cu Effects on APP Protein Expression in The APP/PS1 Transgenic Mouse Brain Lysates

The above supernatants from transgenic mouse brain lysates were also subjected to the SDS-PAGE analysis of Cu effects on APP protein expression. Western blotting was performed on pre-casted NuPAGE™ 4–12% Bis-Tris gels (Invitrogen, Carlsbad, CA, USA) with equal total protein loading (10 μg/well). They were run at 200 V × 45 min and transferred to PVDF membranes at 75 mA/gel × 90 min. The primary antibody was C-terminal rabbit polyclonal antibody for APP (A8717, Sigma-Aldrich, St Louis, MO, USA). The blots were incubated at 1:1000 dilution (in TBST with 10% milk) overnight in the cold room, on a shaker. The blots were then washed for 1-2 h at 15 min intervals. For detection, anti-rabbit antibody at 1:10,000 was added to the blots for 45 min incubation at room temperature on a shaker. The blots were washed for 2-3 h at 30 min intervals, and then developed using a LumiGLO^®^ chemiluminescence kit (KPL Inc., Milford, MA, USA). The final images were captured and analyzed by a VersaDoc™ Digital Imaging System (BIO-RAD, Hercules, CA, USA). For comparison, two Cu(II)-ATPases- APT7B/WND and ATP7A/MNK, and the control protein- β-actin were also probed. The primary antibodies for these proteins were: anti-ATP7B (Novus Biologicals, Littleton, CO, USA), anti-ATP7A (Novus Biologicals, Littleton, CO, USA), and anti-β-actin (A5441, Sigma-Aldrich, St Louis, MO, USA).

### 2.4. ICP-OES Analysis of Dietary Cu Effects on Cerebral Biometal Levels in APP/PS1 Transgenic Mice

To determine the dietary Cu effects on brain biometal levels in APP/PS1 transgenic mice, we analyzed concentrations of biometals (Fe, Cu, and Zn) in cortical lysates from the APP/PS1 transgenic mice using a SPECTRO CIROS VISION ICP-OES spectrometer, which is an ultra-sensitive multi-elemental analysis instrument. The following analytical spectral lines were used: Fe (259.941, 238.204, 239.563 nm); Cu (324.753, 327.393, 224.700 nm); Zn (213.859, 206.201, 202.547 nm), while spectral line of 430.014 nm for Ar was used as a monitoring line as Ar was a carrier gas for analytes. The analytical standard for biometals is CCS-6 (100 µg/mL in 5% HNO3, d = 1.037 g/mL, Inorganic Ventures Inc., Christiansburg, VA, USA). It was diluted in de-ionized water to construct a standard curve with a dynamic range of 10 ppb–500 ppb. To minimize the matrix effects, all lysates and tissue-lysing buffer control were diluted (1:10) in de-ionized water before analysis, and the final concentration values were obtained by averaging over values acquired under different analytical spectral lines and subtracting background biometal concentrations in tissue-lysing buffer. The final concentration values were normalized to total protein concentration in each lysate sample.

### 2.5. Proteomic Analysis of Dietary Cu Effects on Cerebral pro-Inflammatory Cytokines in APP/PS1 Transgenic Mice

#### 2.5.1. Data Collection Using Murine Cytokine Microarray

To quantify the effects of dietary Cu exposure on cerebral pro-inflammatory cytokines in APP/PS1 transgenic mice, the cytokine expression profiles of mouse brain lysate samples were analyzed using the Murine Cytokine Biochip [71,72] from Zyomyx (Hayward, CA, USA). Each Murine Cytokine Biochip measures 30 different mouse cytokines, chemokines and growth factors per sample in a multiplexed assay format. Four chips were run in parallel on the Zyomyx Assay 1200 automated fluidics workstation. Two full chips were used to generate a nine-point calibration curve. Each remaining chip, containing six flow chambers, was used to analyze five lysate samples in parallel, with the remaining chamber used for normalization. Each chamber contained 200 spot features that represented a five-fold replicate for each analyte plus positive and negative controls. A total of 540 analyte data points was obtained per four-chip run.

The protein biochips were first activated and blocked. Then 40 µL of each sample was injected into a flow chamber on a chip for 120-minute incubation at room temperature. After washing, the detection solution was applied for 100 min, followed by a final wash. Upon completion of the multiplexed assays, the protein biochips were scanned on a modified Axon fluorescent scanner, the Zyomyx Scanner 100, equipped with a 532 nm laser. Laser power and PMT voltage were adjusted to provide the largest dynamic range with minimal feature saturation.

#### 2.5.2. Data Analysis for Murine Cytokine Expression

The median feature pixel intensities from the features on each chip were background subtracted after outlier removal using Dixon’s Q-test or boxplot. The background signal was determined from the mean intensity of the negative control features. Feature intensities that exceeded the linear range of the scanner were discarded and the significance above background was determined using a modified Z-factor parameter [73], calculated at a 90% confidence interval. Concentration was calculated using the Zyomyx data reduction software program. Data were averaged and normalized to the calibration standards on each chip. Quantification of each analyte concentration for each sample was calculated using a nine-point multianalyte calibration curve that was derived from the calibration chips. Each reported concentration value represents the average of five replicate measurements from each assay.

## 3. Results

### 3.1. Cu/H_2_O_2_ Exacerbated Aβ Oligomerization In Vitro and Cu Treatment Increased APP Protein Expression Via Its mRNA 5′UTR Translation

As shown in Figure 1, when Aβ1-40 (11.3 µM) was co-incubated with H_2_O_2_ (250 µM) and Cu^2+^ (50 µM) in Tris buffer, pH7.4 for 3 h, apparent Aβ oligomers appeared.

The observed synergistic effect of H_2_O_2_ and Cu upon promoting Aβ oligomerization further corroborates with our previous findings and putative chemical mechanism [74,75]. In addition, as indicated in Figure 2, Cu promoted dose-dependent APP mRNA 5′UTR driven luciferase reporter gene activation in the SH-SY5Y cells.

This may be responsible for increased cellular APP protein expression (Figure 2). This result extends our previous observation- APP mRNA 5′UTR contains an iron-response element [38].

### 3.2. Dietary Cu Exposure Enhanced Cerebral Aβ Amyloid Pathology in APP/PS1 Transgenic Mice

Treatment of Cu for 24 days has not incurred apparent toxicity and behavior disturbances on the mice. This can also be demonstrated by the minimal differences in body weight change between the two groups, as shown in Table 1.

No significant changes in body weight were observed between the regular chow fed and Cu-enhanced chow fed mice. Treatment with Cu for 24 days was associated with changes in levels of metals in the brain. These changes are summarized in Table 2.

Concentrations of copper are elevated in those mice fed the Cu-enhanced Chow. Concentrations of iron and zinc appear to be slightly altered in Cu-fed mice, though these changes are inconsistent and variable, with measured concentrations in the Cu-fed mice that are within or close to the margin of error.

However, Cu treatment was observed to have some effects on cerebral Aβ amyloidosis in APP/PS1 transgenic mice. As shown in Figure 3, the immunostaining for both Aβ1-40 and Aβ1-42 peptides seemed to be stronger in the Cu-treated mouse brain. Interestingly, immunostaining for the astrogliosis marker- GFAP was stronger and broader in Cu-treated mouse brain, although SYP staining was not significantly different in untreated and Cu-treated mouse brain sections.

As indicated in Figure 3, brain extractable Aβ1-40 concentrations showed regional differences in the APP/PS1 transgenic mice. Average cortical Aβ1-40 concentration was 54% higher than striatal one as striatum region is not usually affected by Alzheimer’s amyloid pathology in AD brain [76]. Further, corroborating with the immunostaining result of Aβ1-40, cortical extractable Aβ1-40 concentration increased by 29% in Cu-treated APP/PS1 transgenic mice. In contrast, mean soluble Aβ1-40 concentrations in striatum tissue were not remarkably different in the untreated and Cu-treated transgenic mice.

Since immunohistochemical staining for Aβ peptides was only a qualitative assay, we thus performed the quantitative IHC analysis and ELISA for Aβ levels/concentrations in the brain lysates using commercial kits. The results from the IHC are shown in Table 3.

Cu-enhanced chow fed mice appear to have Aβ1-40 and Aβ1-42 present in larger areas, with more plaque development than the regular chow-fed mice; these differences are within the margin of error, however, so their significance is questionable given the small sample size. The results from ELISA are shown in Table 4.

Compared to the transgenic mice fed on regular chow, Aβ1-42 and total Aβ concentrations increased by 20% and 29%, respectively in Cu-treated APP/PS1 transgenic mice. More interestingly, Aβ1-40 concentration increased considerably (68.7%) relative to Aβ1-42, and correspondingly, both Aβ1-42 percentage and Aβ1-42/Aβ1-40 ratio decreased slightly in the Cu-treated transgenic mice. However, it should be noted that the primary antibodies used are primarily for their respective monomers, and so aggregates are not properly accounted for. Indeed, as aggregation of Aβ1-42 is generally greater than that of Aβ1-40 [77], and Aβ1-42 aggregation is promoted when copper is present, while the aggregation of Aβ1-40 remains relatively unchanged. [78], this may explain the shift in the observed ratio of the primarily monomeric isoforms.

### 3.3. Cu Treatment Increased APP Protein Expression in Both APP/PS1 Transgenic Mouse Brain and Human SH-SY5Y Neuroblastoma cells

In order to analyze the effect of elevated brain-copper levels on APP expression in mice striatum and corticol samples, SDS-PAGE was conducted. β-actin was used as a control, and two copper-transporting ATPases, ATP7B and ATP7A were also used as comparisons. The two transporters are notably copper-efflux transporters, and their expression is regulated by copper levels [79]. The results of the SDS-PAGE are depicted in Figure 4.

Based on the thickness and opacity of the bands, it would appear that APP is expressed in greater quantities in the Cu fed mice; bands were generally lighter in color and opacity (lower expression) in normal mice than the control (β-actin), while APP bands were thicker, with similar opacity to the control. Levels of ATP7B also increased as expected in response to the increase in copper levels, in a manner that is similar to APP. ATP7A levels were lower than those of ATP7B more generally, and though it seems as though its expression is also heightened (more opaque) in copper-fed mice, it is not clear from the gel results alone. The control, β-actin, was fairly consistent between sample groups, as expected. Overall it would seem that increases in levels of copper in APP/PS1 transgenic mouse brain tissue exert a corresponding upregulation on APP expression.

### 3.4. Dietary Cu Exposure Heightened Cerebral pro-Inflammatory Cytokines in APP/PS1 Transgenic Mice

Concentrations of 30 different mouse cytokines, chemokines, and growth factors were determined using a cytokine biochip. Of the thirty molecules surveyed, only two chemokines showed significant differences between the different feeding groups: MCP-5 and TCA4/6Ckine. Their respective concentrations in the different samples are shown in Figure 5.

Both MCP-5 and TCA4/6Ckine exhibited large-magnitude changes associated with the copper-enhanced feeding. Interestingly MCP-5 in particular has been found to be relevant to Alzheimer’s disease. MCP-5 is the murine homologue of human chemoattractant MCP-1 [80]. MCP-1 is also known as CCL2, as it interacts with the CCR2 receptor [81]. TCA4 is a notable ligand of the CCR7 receptor [82].

## 4. Discussion

Our results show support for the idea that excess copper in the brain is associated with various indicators of increased Aβ levels. Our previous study found that copper and hydrogen peroxide interact to promote the oligomerization of Aβ [75]. Our present results corroborate this, showing that Aβ oligomerization occurs in the presence of both Cu^2+^ and H_2_O_2_, but not with Cu^2+^ alone. Aβ oligomerization is thought to be a crucial component of Alzheimer’s disease, as levels of Aβ dimers and other oligomers have been associated with levels of cognitive impairment [83]. One such oligomer in particular, Aβ*56, has been experimentally determined to disrupt memory in young rats [84].

Immunostaining of cortical brain section from the mice revealed an increase in levels of both Aβ1-40 and Aβ1-42 in the copper-fed mice. Further quantitative analysis through ELISA showed increases of the two Aβ forms, though Aβ1-40 levels increased to a much higher, more significant degree; a clear shift in the ratio of Aβ1-40 to Aβ1-42 towards greater Aβ1-40 was noted following exposure to excess copper. Generally, Aβ1-40 accounts for the majority of Aβ isoforms, and this ratio shifts toward a balance of Aβ1-40 and Aβ1-42 in Alzheimer’s cases [85]. Given the metal’s ability to shift the proportion of the isoforms, it is interesting that copper levels in AD patients are not consistent, with conflicting studies indicating excesses and deficiencies in the metal [68], and copper reportedly alleviates amyloid burden in vivo [62,63,64]. However, our assay does not fully identify and quantify Aβ aggregates of either isoform due to the antibody used and, given the increased propensity for Aβ1-42 to aggregate relative to Aβ1-40 [77], this may account for the difference in the observed Aβ1-42:Aβ1-40 ratios between the normal-fed and copper-fed mice. We also note that studies have shown that feeding rats high levels of copper over a period of several weeks to months results in toxic buildups of the metal in the kidney and liver [86,87]. Both the kidney and liver are thought to be involved in the clearance of peripheral Aβ [88,89]. Liver function has been found to be inversely correlated with levels of peripheral Aβ in humans with and without AD [90,91], though current research does not indicate that peripheral levels are correlated with the Aβ1-40 and Aβ1-42 levels of the brain [90].

Our APP gene-luciferase reporter assay confirmed the interaction between Cu and APP gene expression, likely the iron-response element in the 5′ UTR of the APP gene, noted in previous work [38]. Iron response elements typically respond to cellular iron, dissociating from the iron response proteins that bind to them and inhibit translation; thus the presence of iron enables translation [92]. The regulation of copper and iron metabolism and intake/export are linked on both systemic and cellular levels, and the excess of copper due to dietary consumption may correspondingly increase the availability of iron in the SH-SY5Y cells [93,94]. These results were confirmed in vivo, as copper fed mice exhibited increased expression of the APP gene and copper transporter ATP7B relative to the control, β-actin. Results for the second transporter, ATP7A, were somewhat unclear. Further testing with a larger sample size would help elucidate this relationship.

Murine cytokine biochip results revealed copper’s induction of elevated MCP-5 (MCP-1/CCL2 in humans) and TCA4 levels. Notably, CCR2 activity has been found to induce microglial activation and corresponding neuroinflammation in instances of alcohol neurotoxicity and light-induced photoreceptor apoptosis [81,95]. Microglial activation is a key part of neuroinflammation, as the active microglia release more pro-inflammatory cytokines, and the process has been implicated in the development of AD [96,97]. A 2018 two-year follow-up study conducted on AD, MCI, and control subjects found that plasma levels of MCP-1 were significantly higher in AD patients than in individuals with MCI and controls. MCP-1 levels were also highest in those suffering from severe AD [98]. Somewhat conflictingly, a 2006 study found that MCP-1 levels in serum were higher only in those with mild AD or MCI when compared to controls; those with severe AD were not found to be significantly different [99]. Regardless, it seems that MCP-1 has some association with the Alzheimer’s neurogenerative process. CCL2 dominant-negative APP/PS1 mice experienced suppressed astrocyte and microglial responses, improved memory, as well as reduced production and aggregation of Aβ [100]. TCA4 most notably is a chemoattractant involved in T cell homing [101]. CCR7 activity has been implicated in the neuroinflammatory and autoimmune processes of diseases such as multiple sclerosis, but has not yet been directly implicated in AD [102,103]. Despite this, there is evidence that its expression is significantly increased in microglia exposed to Aβ [104]. In a general sense, the neuroinflammation typical to AD is in part consequence of the accumulation of Aβ [105]. However, inflammation also contributes to the expression of APP [106], and the further aggregation of Aβ [105].

## 5. Conclusions

Overall it would seem that copper holds some influence on multiple aspects of amyloidogenesis and inflammation, both of which are significant facets of AD. Though levels of metal in the brains of AD patients are not consistently overabundant or deficient, it is apparent that maintenance of the homeostasis of copper, and other brain biometals more generally, is important to neural health. Our results suggest that metals such as copper have the capacity to promote the formation of potentially harmful Aβ oligomers. Excesses of copper appeared to enhance the expression of APP, which may consequently alter Aβ levels. Though levels of Aβ1-42 in the copper-fed mice were not found to be significantly higher, Aβ1-40 levels exhibited a significant increase in comparison to controls, and the lack of a significant increase in Aβ1-42 may be attributable to the increased aggregation of the peptide. Levels of the murine chemokines MCP-5 and TCA4 were significantly increased; MCP-5 is particularly relevant to the inflammatory responses associated with AD, and the AD neuroinflammation is thought to be both a symptom and propagator of Aβ accumulation. However, the fairly small sample size of our in-vivo experiments poses a considerable weakness of this study, and further corroboration of these results in vivo with larger sample sizes would help to confirm and further expand upon our pilot data.

## Figures and Tables

**Figure 1 biomolecules-10-00408-f001:**
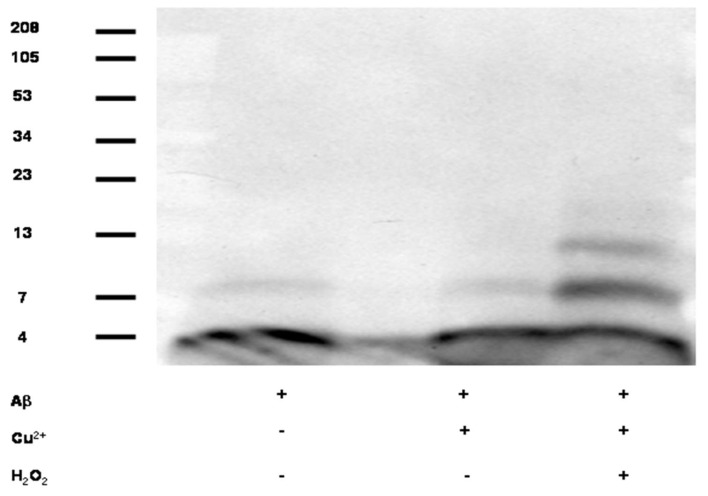
Cu and H_2_O_2_ exacerbated Aβ oligomerization *in vitro*. Western blot represented purified Aβ1-40 (11.3 μM) alone (lane 1) or treated with 50 μM Cu^2+^ (lane 2) or 50 μM Cu^2+^ + 250 μM H_2_O_2_ (lane 3). Bands were detected using an anti-Aβ monoclonal antibody (6E10) that recognizes the N-terminus of Aβ (see Methods for Western blot details).

**Figure 2 biomolecules-10-00408-f002:**
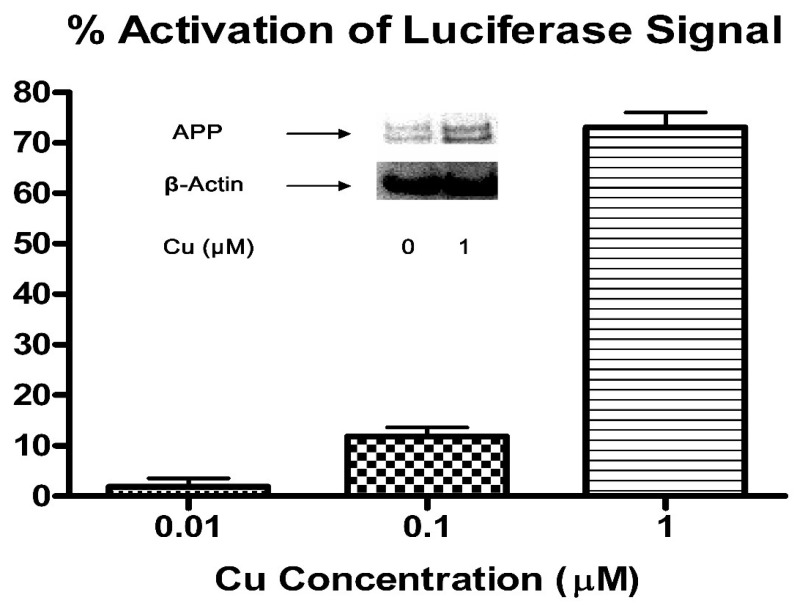
Cu^2+^ treatment increased APP protein expression in human SH-SY5Y neuroblastoma cells via 5′UTR of APP mRNA. Human SH-SY5Y neuroblastoma cells stably transfected with APP 5′UTR driven luciferase reporter gene construct with GFP as an internal specificity control- pIRES(APP 5′UTR ) construct, were treated with Cu^2+^ (0.01, 0.1, 1 µM) for 48 h under regular culture condition (95% O_2_, 5% CO_2_, 95% humidity, 37 °C). And then the luciferase substrate (Perkin Elmer, MA, USA) was added and incubated for 30 minutes for complete cell lysis. Luminescence was recorded on an LJL Analyst plate reader (Molecular Device, CA, USA). Percentage activation of luciferase signal was calculated relative to positive and negative controls. The data indicate mean (±SE, *n* = 5). Inset: Native SH-SY5Y cells were treated with Cu^2+^ (0, 1 µM) under same culture condition for 3 days. Cells were then washed and lysed, and endogenous expression levels of APP and the control protein- β-actin in lysates were probed by Western blotting.

**Figure 3 biomolecules-10-00408-f003:**
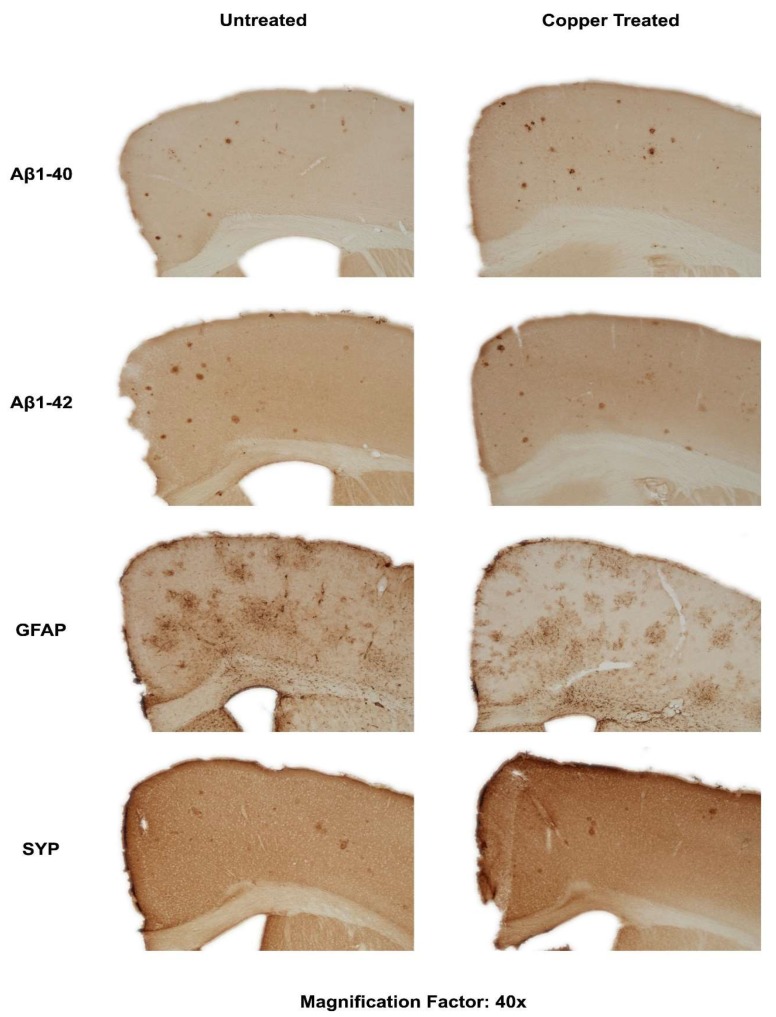
Dietary Cu exposure enhanced cerebral Aβ amyloid pathology in APP/PS1 transgenic mice. Representative coronal brain sections (at the anterior commissure level) from APP/PS1 transgenic mice on regular and Cu-enhanced diets for 24 days. They were immunostained for Aβ1-42 and Aβ1-40, illustrating the size, number and distribution of Aβ amyloid plaques. The immunostaining for reactive astrogliosis marker- GFAP and negative staining for SYP, a marker for neuronal synaptic activity, were also performed.

**Figure 4 biomolecules-10-00408-f004:**
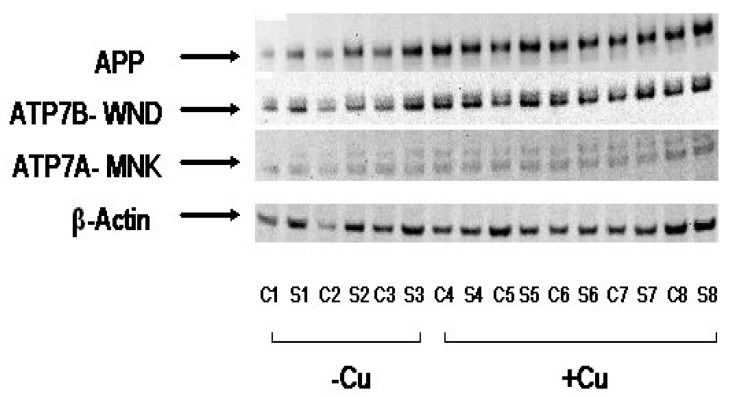
Cu treatment increased APP protein expression in APP/PS1 transgenic mouse brain (A) The protein levels of APP, Cu(II)-ATPases- APT7B/WND and ATP7A/MNK, and the control protein- β-actin in lysates were probed by Western blotting. These cortical (C) and striatum (S) tissue lysates were collected from regular food-fed APP/PS1 transgenic mice (3) and mice fed by Cu-enhanced chow (5).

**Figure 5 biomolecules-10-00408-f005:**
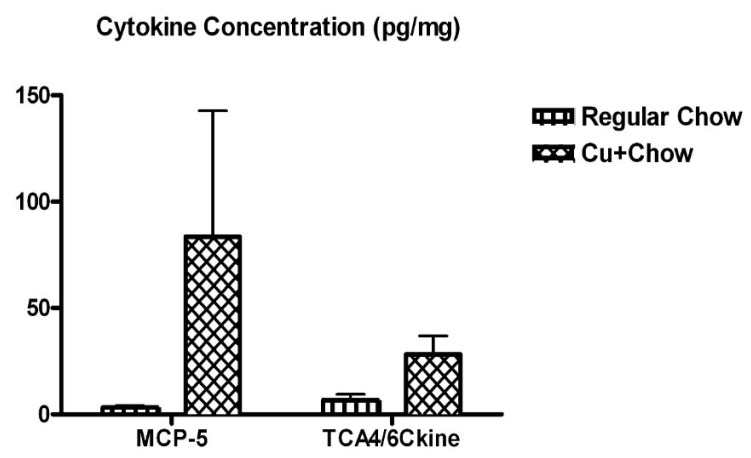
Dietary Cu exposure heightened cerebral pro-inflammatory cytokines in APP/PS1 transgenic mice. 30 different mouse cytokines, chemokines and growth factors in the transgenic mouse cortical lysates were analyzed using Zyomyx Murine Cytokine Biochip System in a multiplexed assay format. Among them, two pro-inflammatory cytokines- MCP-5 and TCA4/6Ckine have shown significant increases in their concentrations due to Cu treatment.

**Table 1 biomolecules-10-00408-t001:** Average Body Weight (g) Changes of the APP/PS1 Transgenic Mice during Cu Treatment Period.

Time Point	Start	Middle	End
Regular chow (*n* = 3)	23.8 ± 0.9	24.2 ± 1.4	23.2 ± 1.0
Cu-enhanced chow (*n* = 5)	24.5 ± 0.8	24.7 ± 0.6	25.1 ± 1.0

* all values are expressed as mean ± SE.

**Table 2 biomolecules-10-00408-t002:** Change of Cortical Cu, Fe, Zn concentrations in the APP/PS1 Transgenic Mice due to Cu Treatment.

Diet	n	Cu (pgmg-1)	Fe (pgmg-1)	Zn (pgmg-1)
Regular Chow	3	25.2 ± 1.3	124.5 ± 25.5	203.0 ± 28.3
Cu-enhanced Chow	5	37.2 ± 2.4	90.3 ± 6.7	177.7 ± 4.6

* all values are expressed as mean ± SE.

**Table 3 biomolecules-10-00408-t003:** Cu-enhanced diets exacerbated brain Aβ amyloidosis in APP/PS1 transgenic mice- IHC analysis results.

Diet	n	Aβ1-40 Area*10^−3^ (µm^2^)	Aβ1-42 Area*10^−3^ (µm^2^)	Aβ1-40 (% of Total Cortex Area Fraction)	Aβ1-42 (% of Total Cortex Area Fraction)	Aβ1-40 Plaque Count	Aβ1-42 Plaque Count	Aβ1-42/Aβ1-40 Area Quotient
Regular Chow	3	118.3 ± 27.4	177.6 ± 27.0	0.52 ± 0.11	0.82 ± 0.09	165 ± 38	193 ± 31	1.50 ± 0.13
Cu-enhanced Chow	5	134.3 ± 12.6	190.3 ± 17.2	0.62 ± 0.07	0.92 ± 0.10	181 ± 22	232 ± 32	1.42 ± 0.07

* All concentrations are normalized to total extracted protein in the cortical tissue lysates, and all values are expressed as mean ± SE.

**Table 4 biomolecules-10-00408-t004:** Cu-enhanced diets exacerbated brain Aβ amyloidosis in APP/PS1 transgenic mice- ELISA results.

Diet	n	Aβ1-40 (pmolg-1)	Aβ1-42 (pmolg-1)	Total Aβ (Aβ1-40 + Aβ1-42) (pmolg-1)	Aβ1-42(% of Total Aβ)	Aβ1-42/Aβ1-40 Quotient
Regular Chow	3	39.31 ± 2.60	15.44 ± 3.42	54.74 ± 6.01	27.40 ± 3.63	0.38 ± 0.07
Cu-enhanced Chow	5	66.32 ± 6.45	18.55 ± 3.80	84.87 ± 9.46	21.14 ± 2.67	0.27 ± 0.04

* All concentrations are normalized to total extracted protein in the cortical tissue lysates, and all values are expressed as mean ± SE.

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
