# Peer review of "A Preliminary Study of Cu Exposure Effects upon Alzheimer’s Amyloid Pathology"

_biomolecules, 2020, doi:10.3390/biom10030408_

Round 1

Reviewer 1 Report

The manuscript titled „In vivo effects of Cu exposure upon Alzheimer’s amyloidogenesis: a pilot study“ aims to unravel the role of copper ions in amyloidogenesis occurring in case of Alzheimer’s disease (AD). To reach this aim authors have explored the effect of copper ions 1) on in vitro oligomerization of amyloid-beta peptide, 2) on expression of GFP reporter gene under the control of the APP mRNA 5’UTR in human SH-SY5Y neuroblastoma cells and 3) on expression of APP, levels of Aβ peptides and cytokines in double transgenic PS1/APP mouse brain.   

The results obtained show that copper ions affect all studied processes in direction, which might intensify amyloidogenesis, characteristic for AD pathology. However, we feel that the work is very preliminary in all three directions, where three different model systems were implemented. Using of different models in one study is justified and encouraged if models are complementary or if the obtained results could be integrated and translated to explain behavior of the modelled system. Unfortunately, it is not the case in the current manuscript, where only fragmented and sometimes debatably new knowledge was obtained, which are only distantly related to AD pathology. This conclusion is based on argumentation, which is presented below.

In the first model the effect of copper ions on in vitro oligomerization of Aβ40 peptide is studied. Oligomerization of Aβ peptides is extremely extensive and complicated field of studies and presenting of one experiment, showing that additional electrophoretic bands appear in HMW region does not add much to the already available enormous amount of information. We have to mention that even this experiment is incompletely described as for example it is even not stated, is it SDS PAGE or other type of electrophoresis. Moreover, the phenomenon that SDS resistant oligomers appear after incubation of Aβ peptides with Cu(II) salts and hydrogen peroxide is known (see for example Williams, T., Serpell, L., Urbanc, B. Stabilization of native amyloid β-protein oligomers by Copper and Hydrogen peroxide Induced Cross-linking of Unmodified Proteins (CHICUP), Biochimica et Biophysica Acta (BBA) - Proteins and Proteomics, 1864(3), 2016, 249-259) and therefore presenting of the result of a single incompletely described experiment does not add much to the field and its publication is unnecessary. Moreover, this experiment does not fit to the manuscript titled “In vivo effects of copper exposure…”

Second model is connected with human SH-SY5Y neuroblastoma cells. Although it is popular cellular model in AD research, it is important to note that these cells are not neurons. The cell culture was originally derived from a metastatic bone tumor biopsy and contains two morphologically distinct phenotypes: neuroblast-like cells and epithelial-like cells. The effects of copper ions on these cancerous cells can not be translated to the neurons or to other types of cells in the human brain. The effect of copper ions on the expression level of GFP reporter gene gene under the control of the APP mRNA 5’UTR or endogeneous APP has been studied, but it is already known that effect of copper ions on APP is more complicated as copper ions affect also distribution and proteolysis of APP, which was not studied by the authors. The effect of copper ions is suggested to occur through interaction with IRE in APP mRNA 5’UTR, which was not confirmed. Slight increase of endogeneous APP expression by the influence of copper ions in SH-SY5Y cell line was documented, however, this result cannot be directly translated into AD context as cellular background of transgenic SH-SY5Y cells is different from brain cells affected in AD. These experiments do also not fit to the manuscript titled “In vivo effects of copper exposure…”

Third, and the only in vivo model, used in the study, is connected with the double transgenic PS1/APP mice. This AD model mice overexpress mutated human APP (isoform 695) with the Swedish mutation (KM670/671NL) and PS1 and begin to accumulate Aβ peptides at a young age and develop large numbers of fibrillar Aβ deposits in the cerebral cortex and hippocampus at about six months of age, considerably earlier than the single transgenic, Tg2576 mice overexpressing only mutated human APP. In the conducted experiment normally feed mice were compared with the mice feed with Cu-enhanced diets, containing a daily dose of approximately 300 mg Cu/kg. Application of such increased copper exposure can hardly model AD, and is rather the model of Wilson disease. This conclusion is supported by the literature data, showing that exposure of rats to 300 mg Cu/kg/day for 6 weeks induces chronic hepatitis (Haywood, S. 1980. The effect of excess dietary copper on the liver and kidney of the male rat. J. Comp. Pathol. 90: 217-232; Haywood, S. 1985. Copper toxicosis and tolerance in the rat. I. Changes in copper content of the liver and kidney. J. Pathol. 145: 149-158). Moreover PS1/APP mice are overexpressing a mutant form of APP and effect of excessive dietary copper on this artificial overexpression system in transgenic animals can not be directly translated to the situation in human AD brain.

To sum it up - we feel that authors present in the manuscript some results from three very preliminary studies using three different model systems, which can hardly be combined into one manuscript titled „In vivo effects of Cu exposure upon Alzheimer’s amyloidogenesis”. This is apparently understood also by the authors, who state in the end of the title that it is: “a pilot study “. We might ask – does the field of AD profit from publication of such a pilot study? Taking into account the facts that in the field of AD there are so many well described evidences for contribution of metal ions (especially copper ions) to the AD pathology, there is no need for further slightly connected preliminary evidences for the putative role of copper ions in AD pathology, but there is rather a need for deeper and complex studies of the metabolism of metal ions in normal and pathological conditions considering cellular systems, animal AD models and ultimately human brain tissue samples. Therefore, we do not support publication of the current manuscript but encourage authors to work further towards such deeper and more comprehensive studies.

Author Response

Reviewer 1’s comments and responses:

The manuscript titled „In vivo effects of Cu exposure upon Alzheimer’s amyloidogenesis: a pilot study“ aims to unravel the role of copper ions in amyloidogenesis occurring in case of Alzheimer’s disease (AD). To reach this aim authors have explored the effect of copper ions 1) on in vitro oligomerization of amyloid-beta peptide, 2) on expression of GFP reporter gene under the control of the APP mRNA 5’UTR in human SH-SY5Y neuroblastoma cells and 3) on expression of APP, levels of Aβ peptides and cytokines in double transgenic PS1/APP mouse brain.   

The results obtained show that copper ions affect all studied processes in direction, which might intensify amyloidogenesis, characteristic for AD pathology. However, we feel that the work is very preliminary in all three directions, where three different model systems were implemented. Using of different models in one study is justified and encouraged if models are complementary or if the obtained results could be integrated and translated to explain behavior of the modelled system. Unfortunately, it is not the case in the current manuscript, where only fragmented and sometimes debatably new knowledge was obtained, which are only distantly related to AD pathology. This conclusion is based on argumentation, which is presented below.

In the first model the effect of copper ions on in vitro oligomerization of Aβ40 peptide is studied. Oligomerization of Aβ peptides is extremely extensive and complicated field of studies and presenting of one experiment, showing that additional electrophoretic bands appear in HMW region does not add much to the already available enormous amount of information. We have to mention that even this experiment is incompletely described as for example it is even not stated, is it SDS PAGE or other type of electrophoresis. Moreover, the phenomenon that SDS resistant oligomers appear after incubation of Aβ peptides with Cu(II) salts and hydrogen peroxide is known (see for example Williams, T., Serpell, L., Urbanc, B. Stabilization of native amyloid β-protein oligomers by Copper and Hydrogen peroxide Induced Cross-linking of Unmodified Proteins (CHICUP), Biochimica et Biophysica Acta (BBA) - Proteins and Proteomics, 1864(3), 2016, 249-259) and therefore presenting of the result of a single incompletely described experiment does not add much to the field and its publication is unnecessary. Moreover, this experiment does not fit to the manuscript titled “In vivo effects of copper exposure…”

Response: This experiment served as an extension of some of our previous work, validating and reinforcing the induction of Aβ oligomer formation by Cu(II) and hydrogen peroxide. SDS-PAGE was used for this experiment, and the incomplete description of this experiment has been rectified (line 131-132). We have changed the title of the manuscript to more accurately describe the content of our work.

Second model is connected with human SH-SY5Y neuroblastoma cells. Although it is popular cellular model in AD research, it is important to note that these cells are not neurons. The cell culture was originally derived from a metastatic bone tumor biopsy and contains two morphologically distinct phenotypes: neuroblast-like cells and epithelial-like cells. The effects of copper ions on these cancerous cells can not be translated to the neurons or to other types of cells in the human brain. The effect of copper ions on the expression level of GFP reporter gene gene under the control of the APP mRNA 5’UTR or endogeneous APP has been studied, but it is already known that effect of copper ions on APP is more complicated as copper ions affect also distribution and proteolysis of APP, which was not studied by the authors. The effect of copper ions is suggested to occur through interaction with IRE in APP mRNA 5’UTR, which was not confirmed. Slight increase of endogeneous APP expression by the influence of copper ions in SH-SY5Y cell line was documented, however, this result cannot be directly translated into AD context as cellular background of transgenic SH-SY5Y cells is different from brain cells affected in AD. These experiments do also not fit to the manuscript titled “In vivo effects of copper exposure…”

Response: We agree that the title does not accurately reflect the body of the work and trying to directly connect the results to in vivo Alzheimer’s Disease pathology is not entirely correct. We have changed the title to more accurately reflect the work. Furthermore, we have changed the wording in the results section for this experiment to avoid relating our in vitro results to the brain/human AD directly.

Third, and the only in vivo model, used in the study, is connected with the double transgenic PS1/APP mice. This AD model mice overexpress mutated human APP (isoform 695) with the Swedish mutation (KM670/671NL) and PS1 and begin to accumulate Aβ peptides at a young age and develop large numbers of fibrillar Aβ deposits in the cerebral cortex and hippocampus at about six months of age, considerably earlier than the single transgenic, Tg2576 mice overexpressing only mutated human APP. In the conducted experiment normally feed mice were compared with the mice feed with Cu-enhanced diets, containing a daily dose of approximately 300 mg Cu/kg. Application of such increased copper exposure can hardly model AD, and is rather the model of Wilson disease. This conclusion is supported by the literature data, showing that exposure of rats to 300 mg Cu/kg/day for 6 weeks induces chronic hepatitis (Haywood, S. 1980. The effect of excess dietary copper on the liver and kidney of the male rat. J. Comp. Pathol. 90: 217-232; Haywood, S. 1985. Copper toxicosis and tolerance in the rat. I. Changes in copper content of the liver and kidney. J. Pathol. 145: 149-158). Moreover PS1/APP mice are overexpressing a mutant form of APP and effect of excessive dietary copper on this artificial overexpression system in transgenic animals can not be directly translated to the situation in human AD brain.

Response: While we agree that the kidney and liver toxicity of copper should be mentioned, it seems the references you provided involve dietary copper treatment at doses considerably higher than 300mg/kg, over a longer period of time than our study (24 days). We observed that the health and general wellbeing (like body weight change) of the mice utilized in our study was stable overall over the duration of our experiment. Nonetheless, we have added an explanation of copper’s liver & kidney toxicity, using the references you provided, and the relevance to peripheral Aβ clearance and brain Aβ levels (line 468-474). However, we agree that insinuating that our models directly connect to the Alzheimer’s Disease condition in the title is incorrect, and this has been changed. We also agree that no better AD mouse model can truly simulate the pathology in human AD brain, and this is the inherent experimental deficiency.

To sum it up - we feel that authors present in the manuscript some results from three very preliminary studies using three different model systems, which can hardly be combined into one manuscript titled „In vivo effects of Cu exposure upon Alzheimer’s amyloidogenesis”. This is apparently understood also by the authors, who state in the end of the title that it is: “a pilot study “. We might ask – does the field of AD profit from publication of such a pilot study? Taking into account the facts that in the field of AD there are so many well described evidences for contribution of metal ions (especially copper ions) to the AD pathology, there is no need for further slightly connected preliminary evidences for the putative role of copper ions in AD pathology, but there is rather a need for deeper and complex studies of the metabolism of metal ions in normal and pathological conditions considering cellular systems, animal AD models and ultimately human brain tissue samples. Therefore, we do not support publication of the current manuscript but encourage authors to work further towards such deeper and more comprehensive studies.

Response: We appreciate the reviewer’s critiques. We need to point out that we have only validated findings from our and other groups on the exacerbation of Aβ oligomerization by Cu and H2O2 in vitro. However, this is a first study on Cu exposure and AD using our particular APP/PS1 AD transgenic model (it is different from Jackson Lab APP/PS1 mouse model). It may only represent a validation and incremental advancement in our understanding of AD metallobiology. However, we believe it still deserves a publication in Biomolecues as the pilot data reported in this manuscript will be used for planning deeper and more comprehensive studies in the near future.

Reviewer 2 Report

The study by Pilozzi et al concerns the in vivo effects of Cu exposure in relation to amyloid beta aggregation and amyloid formation. This is an important topic, and in my opinion the study deserves to be published. I do however have some minor concerns.

  1. Figure 1 shows the effect of Cu and H2O2 on Abeta oligomerization, arguably in combination with dityrosine formation. Three conditions are shown: Abeta alone, Abeta + Cu, and Abeta + Cu + H2O2. The last condition shows the strongest effect. From these data, one could conclude that Cu does nothing and that the oligomerization is induced by H2O2. Most likely though, the combination of Cu and H2O2 is required. To demonstrate this, a control experiment of Abeta with H2O2 but without Cu is required. Furthermore, I recommend the authors to read the article "A central role for dityrosine crosslinking of Amyloid-beta in Alzheimer's disease, Acta Neuropathol. Commun. 1 (2013) 83", and possibly include it as a reference.
  2. In Table 3B the authors show the levels of AB40 and AB42, and discuss how these levels change after Cu exposure. Here, and in the discussion of these data, it is very important to point out that the reported numbers only show observed AB levels, but NOT total AB levels. Both AB40 and AB42 levels increase as a result of Cu exposure. But the AB42 peptide is very prone to aggregate, and thus will exist both in monomeric and aggregated form. The authors use ELISA to measure the AB levels, with an antibody that mainly detects AB monomers. Thus, the AB42 aggregates are not detected (nor the AB40 aggregates, but there are probably fewer of those). The statement in the discussion saying that Cu is able to "shift the proportion of the isoforms", is possibly incorrect. At least, such a conclusion can not be drawn from the current data. That said, the observation that Cu exposure increases AB production is interesting enough (although not completely new).
  3. In the introduction, it is stated that "trace amounts of copper in water induce AB amyloid plaques and learning deficits [49,50]". Here, it might be relevant to separate between studies on animals and humans. Furthermore, the effect of Cu exposure on human populations remains debated - see for example the article "Low copper-2 intake in Switzerland does not result in lower incidence of Alzheimer’s disease and contradicts the Copper-2 Hypothesis, Chemical Research in Toxicology 2020, DOI: 10.1177/1535370219899898". The Cu effects appear to be more established at a molecular level - see for example the recent review "Metal binding to the amyloid‑β peptides in the presence of biomembranes: potential mechanisms of cell toxicity, Journal of Biological Inorganic Chemistry (2019) 24:1189–1196". In this context, the current study with detailed results from animal models is very welcome.
  4. The authors focus their discussion on the role of biometals such as Cu, Fe, and Zn in AD. It might be worth mentioning somewhere in the introduction or discussion that exogenous heavy metals such as lead and mercury might also be relevant - see for example the articles "Mercury and Alzheimer’s Disease: Hg(II) Ions Display Specific Binding to the Amyloid- Peptide and Hinder Its Fibrillization, Biomolecules 2020, 10, 44; doi:10.3390/biom10010044" and "Alzheimer’s disease and cigarette smoke components: effects of nicotine, PAHs, and Cd(II), Cr(III), Pb(II), Pb(IV) ions on amyloid-β peptide aggregation, Scientific Reports 7: 14423, DOI:10.1038/s41598-017-13759-5".

Author Response

Reviewer 2’s comments and responses:

The study by Pilozzi et al concerns the in vivo effects of Cu exposure in relation to amyloid beta aggregation and amyloid formation. This is an important topic, and in my opinion the study deserves to be published. I do however have some minor concerns.

  • Figure 1 shows the effect of Cu and H2O2 on Abeta oligomerization, arguably in combination with dityrosine formation. Three conditions are shown: Abeta alone, Abeta + Cu, and Abeta + Cu + H2O2. The last condition shows the strongest effect. From these data, one could conclude that Cu does nothing and that the oligomerization is induced by H2O2. Most likely though, the combination of Cu and H2O2 is required. To demonstrate this, a control experiment of Abeta with H2O2 but without Cu is required. Furthermore, I recommend the authors to read the article "A central role for dityrosine crosslinking of Amyloid-beta in Alzheimer's disease, Acta Neuropathol. Commun. 1 (2013) 83", and possibly include it as a reference.

Response: The H2O2 experiment was for reinforcement of one of our previous works. Although you are correct that a trial with only H2O2 would be beneficial, but our previous work indicates there are always a trace amounts of biometals including Cu in the experimental buffer (Huang et al, Journal of Biological Inorganic Chemistry 2004;9:954-60) . That is why we did not consider Abeta with only H2O2 but without Cu would be a valid condition. We had mentioned the dityrosine crosslinking mechanism, as it was a subject of one of our previous works. We have added the reference you mentioned as additional reinforcement; thank you for providing it.

  • In Table 3B the authors show the levels of AB40 and AB42, and discuss how these levels change after Cu exposure. Here, and in the discussion of these data, it is very important to point out that the reported numbers only show observed AB levels, but NOT total AB levels. Both AB40 and AB42 levels increase as a result of Cu exposure. But the AB42 peptide is very prone to aggregate, and thus will exist both in monomeric and aggregated form. The authors use ELISA to measure the AB levels, with an antibody that mainly detects AB monomers. Thus, the AB42 aggregates are not detected (nor the AB40 aggregates, but there are probably fewer of those). The statement in the discussion saying that Cu is able to "shift the proportion of the isoforms", is possibly incorrect. At least, such a conclusion can not be drawn from the current data. That said, the observation that Cu exposure increases AB production is interesting enough (although not completely new).”

Response: You are correct in that disproportional AB42 aggregation relative to that of AB40 may explain the observed shift in the ratio of monomeric isoforms, and we have made a note of this in the text.

  • In the introduction, it is stated that "trace amounts of copper in water induce AB amyloid plaques and learning deficits [49,50]". Here, it might be relevant to separate between studies on animals and humans.” Furthermore, the effect of Cu exposure on human populations remains debated - see for example the article " Low copper-2 intake in Switzerland does not result in lower incidence of Alzheimer’s disease and contradicts the Copper-2 Hypothesis, Chemical Research in Toxicology 2020, DOI: 10.1177/1535370219899898". The Cu effects appear to be more established at a molecular level - see for example the recent review "Metal binding to the amyloid‑β peptides in the presence of biomembranes: potential mechanisms of cell toxicity, Journal of Biological Inorganic Chemistry (2019) 24:1189–1196". In this context, the current study with detailed results from animal models is very welcome.”

Response: We have separated the results from animal and human experimentation/observation as suggested. Additionally, we have included the selected reference in noting the conflicting evidence for and against the copper/AD hypothesis. We thank you for providing the reference.

  • The authors focus their discussion on the role of biometals such as Cu, Fe, and Zn in AD. It might be worth mentioning somewhere in the introduction or discussion that exogenous heavy metals such as lead and mercury might also be relevant - see for example the articles "Mercury and Alzheimer’s Disease: Hg(II) Ions Display Specific Binding to the Amyloid Peptide and Hinder Its Fibrillization, Biomolecules 2020, 10, 44; doi:10.3390/biom10010044" and "Alzheimer’s disease and cigarette smoke components: effects of nicotine, PAHs, and Cd(II), Cr(III), Pb(II), Pb(IV) ions on amyloid-β peptide aggregation, Scientific Reports 7: 14423, DOI:10.1038/s41598-017-13759-5".

Response: We have briefly mentioned the relevance of other metals to Alzheimer’s disease, and have included the references you provided, among others. Thank you for providing them.

Reviewer 3 Report

1 The instruction section seems too long, the authors should shorten.

2 For the animal studies, the sample size is relatively small, therefore the explanation of results require cautions. Also, what is the difference for PS1/APP mice and the widely used commercially available APP/PS1 mice from Jackson laboratory?

Author Response

Reviewer 3’s comments and responses:

  1. The instruction section seems too long, the authors should shorten.

Response: We have condensed and removed certain sections of the introduction to shorten it.

  1. For the animal studies, the sample size is relatively small, therefore the explanation of results require cautions. Also, what is the difference for PS1/APP mice and the widely used commercially available APP/PS1 mice from Jackson laboratory?

Response: You are correct that the sample sizes for our in vivo experiments were small, and we have made note of this in the text. The mice we used were cross-bred from mutant (M146V) PS-1 knock-in mice and APPTg2576 mice- i.e. Swedish mutation (Mo/HuAPP695swe). However, the mice from Jackson Laboratory express a mouse-human chimeric APP with the Swedish mutation (Mo/HuAPP695swe) and mutant PS1 (PS1-dE9).

Round 2

Reviewer 1 Report

The phenomenon that SDS resistant oligomers appear after Aβ40 incubation with Cu(II) salts and hydrogen peroxide is not new. Besides the preliminary data from the authors (Ref. 75 - Huang, X.; Moir, R.D.; Tanzi, R.E.; Bush, A.I.; Rogers, J.T. Redox-active metals, oxidative stress, and Alzheimer's disease pathology. Ann N Y Acad Sci 2004, 1012, 153-163) there are much more relevant papers, which should be cited and also discussed to show novelty of the obtained result. One recent paper has also been mentioned in the earlier review (Thomas L. Williams, Louise C. Serpell, Brigita Urbanc, Stabilization of native amyloid β-protein oligomers by Copper and Hydrogen peroxide Induced Cross-linking of Unmodified Proteins (CHICUP), Biochimica et Biophysica Acta (BBA) - Proteins and Proteomics, Volume 1864, Issue 3, 2016, Pages 249-259) and we actually do not understand, why this comment was not considered by the authors. 

Reviewer 2 Report

With the changes made in this revised version I am happy with the manuscript, and I think it is ready to be published.

One small comment about the response to my thoughts about the dityrosine crosslinking experiments: if the authors conduct such measurements in future work, one way to obtain a control experiment with H2O2 but without any metals present, and also in only buffer without any metals present, is to conduct the experiments in the presence of EDTA. It would however be unnecessary to ask for such detailed control experiments in the current study, where the main focus is not on cross-linking effects.

Reviewer 3 Report

Authors have responded to all the questions. I would suggest the manuscript will be accepted!